# Physical Activity Patterns of Women with a Twin Pregnancy—A Cross-Sectional Study

**DOI:** 10.3390/ijerph18157724

**Published:** 2021-07-21

**Authors:** Katarzyna Kwiatkowska, Katarzyna Kosińska-Kaczyńska, Izabela Walasik, Agnieszka Osińska, Iwona Szymusik

**Affiliations:** 1Students Scientific Association, 1st Department of Obstetrics and Gynecology, First Faculty of Medicine, Medical University of Warsaw, 02-015 Warsaw, Poland; katarzyna.kwiatkowska1088@gmail.com (K.K.); izabela.a.walasik@gmail.com (I.W.); 2The Center of Postgraduate Medical Education, 2nd Department of Obstetrics and Gynecology, 01-813 Warsaw, Poland; a_osinska@interia.pl; 31st Department of Obstetrics and Gynecology, First Faculty of Medicine, Medical University of Warsaw, 02-015 Warsaw, Poland; iwona.szymusik@wum.edu.pl

**Keywords:** twin pregnancy, physical activity, preterm delivery, pregnancy

## Abstract

Background: No specific physical activity guidelines are available for women in multiple pregnancy. Aim of the study was to assess the knowledge and experience of women regarding physical activity during their latest twin pregnancy. Methods: A cross-sectional study including women after a twin delivery was conducted in Poland. A questionnaire was distributed in 2018 via web pages and Facebook groups designed for pregnant women. Results: 652 women filled out the questionnaire completely. Only 25% of women performed any physical exercises during twin gestation. The frequency of preterm delivery was similar in physically active and non-active participants. 35% of the respondents claimed to have gained information on proper activity from obstetricians during antenatal counselling while 11% claimed to be unable to identify the reliable sources of information. 7% of women admitted to feel discriminated by social opinion on exercising during a twin pregnancy. Conclusions: The population of women with a twin gestation is not sufficiently physically active and is often discouraged from performing exercises during gestation. Therefore, it is crucial to inform obstetricians to recommend active lifestyle during a twin gestation and to provide reliable information on physical activity to pregnant women. Further research on this topic is necessary in order for obstetric providers to counsel women on appropriate exercise with a twin pregnancy.

## 1. Introduction

Physical activity during pregnancy is an essential element of a healthy lifestyle. Regular aerobic exercise during pregnancy has been shown to decrease several pregnancy complications such as gestational diabetes mellitus or gestational hypertension occurrence, preterm birth, cesarean birth rate and to decrease postpartum recovery time in singleton gestation [1,2,3,4]. Physical activity in essential in prevention of depressive disorders in postpartum period [5]. Obstetricians and other health care providers should encourage their patients to continue or to commence exercising as an important component of optimal health [6]. The American College of Obstetricians and Gynecologists (ACOG) created recommendations on physical activity during gestation and encouraged all pregnant women to engage in moderate-intensity exercise consisting of 150 min per week [6]. The World Health Organization (WHO) recommends pregnant women to do at least 150 min of moderate-intensity aerobic physical activity throughout the week, incorporate a variety of aerobic and muscle-strengthening activities and limit sedentary time [7]. The above guidelines concern all pregnant women. However, due to the lack of data on physical activity in multiple gestation, multiples are not mentioned separately. They are mentioned in Canadian guidelines on physical activity, however according to it twin pregnancy beyond 28 weeks of gestation is a relative contraindication to perform exercises [8]. Moreover, no specific physical activity guidelines are available for women with a multiple pregnancy.

Twin birth rate is on the rise. Increasing trends in twin birth rates have been observed worldwide; in the USA, the rate of twin deliveries has increased by almost 80% since the 1980s and in Poland an increase of 27% of multiple birth rate has been observed since 1997 [9,10,11]. In 2018 a total of 3% of all births were multiple in Poland [12]. Compared to singleton neonates, twins are at an up to seven-fold higher risk of perinatal mortality and morbidity, which are strongly related to prematurity [13]. Infants are also more prone to perinatal morbidity and mortality compared to singletons, due to complications related to monochorionicity, intrauterine growth restriction or preterm birth. Over two-thirds of twins are born preterm which contributes to 50% of all neonatal mortality [14]. Numerous non-modifiable risk factors of pregnancy complications may be listed, including genetic or immune predispositions for specific pathologies. However, other risk factors are modifiable. Lifestyle, proper nutrition and physical activity may lower the risk of pregnancy complications and have an impact on perinatal outcomes in singletons [6,15]. As multiple pregnancy is at an especially high risk of complications, it is essential for health care providers to evaluate the risks and benefits of physical activity in pregnant women to counsel them properly in order to lower the risk of adverse perinatal outcome. We hypothesize that, analogously to a singleton gestation, only a part of women performs exercises during twin pregnancy although physical activity has no adverse impact on pregnancy outcome. The objective of this study was to: assess the patterns of respondents’ physical activity during their latest twin pregnancy, their sources of information and experience regarding physical activity and its relation to the course of pregnancy and delivery.

## 2. Materials and Methods

### 2.1. Setting

It was a cross-sectional study including women after a twin delivery conducted in Poland. A questionnaire in the Polish language was invented by the authors of the study and distributed between November and December 2018 via web pages and Facebook groups designed for pregnant women which already existed. Groups were not correlated with some specific topic and has been randomly elected. Web pages written in the Polish language provided general medical and social information on the pregnancy and care of a baby and were available without any limits and Facebook groups were accessible for women declaring to be pregnant or young mothers without any special requirements. The questionnaire was dedicated to Polish-speaking women, regardless of inhabitancy, who delivered twins within the last year before the survey. The questionnaire was administered by Google Forms (Google LLC, Mountain View, CA, USA). In addition, the link to it was presented on Facebook groups and web pages designed for pregnant women. The survey was voluntary and anonymous—no questions regarding personal data that would enable the identification of participants were included and only the authors of the study had access to the collected information. Access to the questionnaire was granted after login in only once.

### 2.2. The Questionnaire

The questionnaire used was self-composed by the authors and had not been previously validated and the veracity of given answers was impossible to verify, which could induce bias to the study. The questionnaire consisted of single or multiple-choice closed questions which evaluated the knowledge and experience related to physical activity during a twin pregnancy. The survey was divided into five sections depending on which aspects they concerned. The first part consisted of sociodemographic and personal data. The respondents were asked about their age, education, inhabitancy, weight and height, parity and physical activity before the pregnancy. In the second part women were asked about specific recommendations given to them regarding exercise during pregnancy and where they looked for information on physical activity during gestation. The third part concerned physical activity patterns during pregnancy, including the type of exercise, frequency and duration of training per week in each trimester of pregnancy. The fourth part referred to information about most common pregnancy ailments (constipation, leg swelling, back pain, mood swing, fatigue, sleep problems, decreased libido, heartburn, leg crumps or varicose veins occurrence) and the course of pregnancy and delivery, including gestational weight gain, gestational hypertension, preeclampsia, intrahepatic cholestasis of pregnancy or gestational diabetes mellitus occurrence, gestational age at delivery and neonatal birthweight. The list of pregnancy ailments was created on the basis of previously published research [16]. The fifth one investigated the women’s opinion on family and social attitude to exercising during pregnancy. The questionnaire is presented in the Appendix A.

### 2.3. Sampling

The respondents were qualified as “physically active during pregnancy” if they performed exercises such as regular walks, marching, jogging, total body workout at a gym, swimming, yoga, pilates, fitness, exercise-ball workouts or home gymnastics. Exercises had to be carried out regularly (at least twice a week) and one training should be at least 15 min long. Women who did not perform any type of the above-mentioned activities regularly or did not exercise at all were qualified as “physically non-active”. Previously, we conducted and published a similar study on singleton gestations [17].

The inclusion criteria were age above 18 years old, delivery at or beyond 24 weeks of gestation maximum one year prior to completing the questionnaire and a live twin birth. The exclusion criteria were as follows: miscarriage, any major disabilities and monochorionic monoamniotic pregnancies. Only questionnaires which were completely filled out were taken into account. All the answers were checked for duplicates and no identical records were found.

### 2.4. Variables

Body mass index (BMI) was defined as the woman’s weight in kilograms divided by her height in meters squared. The following ranges were considered: underweight: <18.5 kg/m^2^; normal range: 18.5–24.99 kg/m^2^; overweight: 25–29.99 kg/m^2^; obese: >30 kg/m^2^. Gestational weight gain (GWG) was defined as the difference between maternal weight at delivery and preconception weight. For twin pregnancies, the range of weight gain throughout a pregnancy recommended by the Institute of Medicine for normal weighing women was 16.8–24.5 kg, for overweight women 14.1–22.7 kg and for obese women 11.3–19.1 kg [18]. Preterm delivery (PTD) was defined as one occurring before 37 weeks of gestation. Neonatal weight below 2500 g was considered as low birth weight. Neonatal birth weight was classified to a specific percentile according to twin growth charts [19]. Pre-gestational weight, height, weight at delivery, type of twin pregnancy chorionicity, gestational age at delivery and neonatal birthweight data were obtained from the questionnaire.

### 2.5. Statistical Analysis

Survey data were analyzed using R version 3.2.5 (R Foundation for Statistical Computing, Vienna, Austria); the χ^2^ or Fisher exact tests were used to compare categorical variables and the Mann-Whitney U-test was used for continuous variables with *p* ≤ 0.05 being considered significant. The data were reported as absolute numbers and percentages or means and standard deviations. Respondents were divided into two groups of physically active and non-active women and those groups were compared.

### 2.6. Ethic Committee Approval

The study protocol obtained the approval of the Ethics Committee of the Medical University of Warsaw (AKBE/293/2019). The committee waived the obligation to gain a written or verbal consent in order to participate in the study as fulfilling the questionnaire was tantamount to giving consent.

## 3. Results

### 3.1. Study Population

652 women filled out the questionnaire completely. 74% delivered dichorionic diamniotic twins and 26%—monochorionic diamniotic twins. The maternal characteristics of the study group are presented in Table 1.

### 3.2. Physical Activity during Pregnancy

The study group was further divided into physically active and non-active respondents during a twin gestation. Only 25% of women with a twin pregnancy performed any physical exercises during gestation. The baseline characteristics of physically active and non-active respondents are presented in Table 1. Only 7.2% of women exercised for >150 min per week in the first, 5.2% in the second and 2.1% in the third trimester of pregnancy.

The most common activities performed during a twin gestation were: 64%—walking, 19%—swimming, 19%—home gymnastics, 18%—exercise ball workouts, 16%—marching/brisk walking, 11%—total body workout at the gym and 6%—running.

In the first trimester of a twin pregnancy 24% of the women claimed to perform any kind of exercise. The frequency and duration of exercises in three trimesters of pregnancy are presented in Figure 1 and Figure 2. The most common frequency of physical activity was at least 5 times per week (43%). Active respondents chose activities such as walking (72%), home gymnastics (30%), swimming (28%) or exercise-ball workouts (19%). However, the total declared duration of training per week in the first trimester usually was shorter than 60 min (47%).

A similar percentage of women exercised in the second trimester (22% vs. 24% *p* = 0.5). Comparable rates of respondents exercised 1–2 (32%), 3–4 (34%) or at least 5 times a week (34%) and the usual duration of exercising was below 60 min weekly (52%). The respondents performed activities such as walking (85%), home gymnastics (25%), swimming (25%) or fitness (18%).

Significantly fewer women exercised during the third trimester of pregnancy (first trimester vs. third: 24% vs. 14%; *p* = 0.001). They usually performed exercises 3–4 times a week (34%). The number of respondents declaring to exercise at least 5 times a week was significantly lower than in the first trimester (43% vs. 27%; *p* < 0.05). Pregnant women preferred activities such as walking (87%), marching (18%) or exercise ball workouts (22%). Significantly more women declared to exercise less than 60 min per week than in the first (71% vs. 47%; *p* < 0.01) and in the second trimester of pregnancy (vs. 52%; *p* < 0.004). The rates of respondents declaring to be physically active from 60 to 150 min weekly was similar in all three trimesters, while the percentage of women declaring to exercise over 150 min per week decreased in the following trimesters (29%, 21% and 9%, respectively; *p* < 0.02). Walking was the most popular form of physical activity throughout the twin pregnancy.

The respondents claimed to choose to be physically active during a twin pregnancy to: improve the overall fitness (73%), continue pre-pregnancy physical activity (51%), enable faster recovery after the forthcoming delivery (43%) and to prepare herself for delivery (9%). 21% of active women exercised regularly until the delivery date and 9% to the last week before labor. However, most of the respondents stopped exercising in the second (24%) or third trimester (33%).

A total of 75% of women did not perform any kind of exercises during a twin gestation. The main reasons for the lack of physical activity were medical contraindications determined by a health care provider (51%), feeling of the lack of energy (44%), fear for the infant’s health (36%), the lack of will (27%) and the lack of knowledge regarding suitable exercises (18%).

### 3.3. Pregnancy Ailments

We investigated the occurrence of the most common complaints related to pregnancy and found that back pain, fatigue and sleeping problems occurred statistically less commonly in the group of exercising respondents (Table 2).

### 3.4. The Course of Pregnancy and Delivery

Table 3 presents associations between exercising, the course of pregnancy, delivery and the newborns’ birth weight according to data provided by the respondents. There were no preterm deliveries prior to 30 weeks of gestation in the physically active group (0% vs. 5%; *p* = 0.002). The frequency of PTD was similar in physically active and non-active groups. GDM occurred significantly more commonly in women exercising during pregnancy (41% vs. 29%; *p* = 0.04), while no such relation was observed regarding any other analyzed pregnancy complications. No relation between physical activity and GWG was observed.

### 3.5. The Sources of Information about Physical Activity during Pregnancy

Women were asked about the sources of their knowledge on proper physical activity during pregnancy. Only one third of the respondents claimed to have gained information from obstetricians during antenatal counselling (35%). 61% of women sought out information on the internet, 26% in books, 25% in childbirth school meetings or from friends, 16% had physiotherapist advice and only 14% received information from midwives. 11% of all the respondents claimed to be unable to identify the reliable sources of information regarding exercise during pregnancy.

### 3.6. Social Attitude toward Physical Activity during Pregnancy

Family and social attitude towards physically active pregnant women was assessed in the respondents’ subjective view in a subgroup of active women. 70% of the respondents claimed to have family and partner support in performing physical activity during gestation. 32% met with a positive reaction from the others. Conversely, 7% of women admitted feeling discriminated by social opinion on exercising during a twin pregnancy. 2% of the respondents performed exercises away from their homes to avoid both being recognized and social discrimination. 19% admitted to hearing negative opinions about their physical activity during pregnancy, while 32% of women met with positive reactions of society. The lack of acceptance appeared even among family members (7%).

## 4. Discussion

It is the first study investigating the patterns of physical activity in a large group of women with a twin gestation. We found that only 25% of women with a twin pregnancy performed any physical exercises during gestation compared to 52% of women with a singleton pregnancy, which we found in our previous study [17]. The frequency of exercises decreased along the pregnancy, while the usual declared total time of physical activity did not exceed 60 min per week in any trimester. Physically active women did not deliver preterm more often than non-active ones. As many as 19% of the respondents admitted to hearing negative opinions about their physical activity during pregnancy.

In our study 75% of the respondents did not perform any activities during pregnancy and 46% of non-active women did not exercise before the pregnancy either. No data are available on the rates of physically active women with a twin gestation in the literature. In a singleton population 60% on women who did not exercise during pregnancy were physically non-active before the gestation as well [17]. According to our data women with a twin gestation performed any physical activities less often than women with a singleton gestation. In our previously published paper 52% of the respondents with a singleton pregnancy claimed to perform regular exercises during gestation [17]. Similar results were presented by Evenson et al., who performed telephone interviews. They acquired data from 1979 pregnant women and found that the prevalence of any leisure activity in the past month was 66% in pregnant women [20]. In the NHANES study 54% of the participants reported any moderate to vigorous household activity and 57% reported any moderate to vigorous leisure activity during pregnancy [21]. In our study 51% of the respondents claimed to have contraindications to exercises determined by a health care provider. According to Whitaker et al., 42% of pregnant women in the USA were recommended to restrict their activity levels with additional 13% being prescribed bed rest [11]. Bed rest is commonly advised to prevent preterm delivery. However, no evidence is available on its efficacy to decrease PTD in singletons or twins [22,23]. da Silva Lopes et al., conducted a systematic review and assessed the effectiveness of bed rest in hospital or at home to improve perinatal outcomes in women with a multiple pregnancy. The review included six trials, involving a total of 636 women with a twin or triplet pregnancy. Bed rest did not reduce the risk of very preterm birth (risk ratio (RR) 1.02, 95% confidence interval (CI) 0.66 to 1.58), perinatal mortality (RR 0.65, 95% CI 0.35 to 1.21) and low birth weight (RR 0.95, 95% CI 0.75 to 1.21) in multiples [24]. We found no differences in PTD occurrence between physically active and non-active women in our survey and a lower incidence of delivery prior to 30 weeks in the active group of respondents. The restriction of activity increased the risk of thromboembolic events, bone loss, muscle weakness, lower birth weight of newborns and depression and anxiety [25,26,27,28,29,30]. As the rate of women performing physical activities during a twin gestation is so low, it is necessary to provide suitable advice on physical activity to pregnant women in Poland.

We found GDM to be more common in the physically active group of respondents. We hypothesize that women with GDM are more frequently advised to perform regular psychical exercises in order to obtain normoglycemia. The Polish Society of Obstetricians and Gynecologists recommends maintaining moderate physical activity throughout the pregnancy in women with GDM [31]. No other associations between physical activity and pregnancy complications were noted.

Approximately 35% of study participants reported to have gained advice on physical activity during their twin pregnancy from an obstetrician. In a cross-sectional electronic survey Whitaker et al., analyzed the opinions of 276 women pregnant with twins and found that 63% of the women reported to have gained advice on physical activity and nutrition from a health care professional [11]. In our study only 35% of women gained information from obstetricians during antenatal counselling. According to Findley et al., almost 50% of study participants reported that the advice they received from gynecologists on activity was unclear and/or conflicting in nature. Most participants often sought information for themselves by searching the internet, which, again, often led to varying/contradicting information which was liable to be misunderstood [32]. According to our results the internet was the most popular source of information regarding physical activity in pregnancy with 61% of women using it to seek out information. However, every tenth woman claimed to be unable to identify any reliable sources of information regarding exercise during pregnancy.

We did not observe any significant relations between physical activity during a twin pregnancy and vaginal delivery rate. Such a correlation was observed in singletons. Physically active women had a vaginal birth (61% vs. 55%; *p* < 0.001) and were more likely to have a spontaneous onset of the delivery compared to non-active women (74% vs. 71%; *p* = 0.001) [17]. Bakarat et al., conducted a randomized controlled trial of 290 healthy pregnant Spanish women with a singleton gestation. The percentage of cesarean and instrumental deliveries in the physically active group was lower than in the non-active group (16%, 12%, vs. 23%, 19%, respectively; *p* = 0.03). Based on those results, a supervised program of moderate-intensity exercise performed throughout pregnancy was associated with a reduction in the rate of cesarean section [33]. The lack of such a relationship between vaginal delivery and physical activity during pregnancy in twin gestation in Poland may be due to a high rate of cesarean deliveries in twins. In our study 93% of all respondents delivered via cesarean section. It may be due to a former recommendation of the Polish Society of Obstetricians and Gynecologists, which classified monochorionic twin gestation and ART pregnancy as a relative indication for operative delivery regardless of the position of the fetuses. As most women with a twin gestation in our survey had cesarean delivery, the relationship between physical activity and successful vaginal birth may be difficult to determine.

It is one of the first studies evaluating physical activity patterns among women pregnant with twins. No survey conducted on such a large population of women has been published. The survey was anonymous and distributed via the internet which promotes honest answers. The sample of respondents was diverse regarding sociodemographic characteristics such as age, education, inhabitancy and the number of deliveries. However, there are several limitations of the study. In the survey women self-reported their physical activity. As respondents up to one year after delivery were included to the study, the accuracy of recalling information might decrease over time. All the answers were subjective and impossible to be verified by the research group. It is possible that people may tend to provide answers which create a better view of themselves than the true one and may claim to be more active than they really were. It especially underlines the importance of the problem as only one-fourth of the respondents claimed to perform any kind of physical activities during gestation. Another limitation is related to the fact that the questionnaire used in our study was self-composed by the authors and had not been previously validated in a pregnant population. Even though the survey was available without any special requirements and limitations except of inclusion criteria, the study group consisted of women using internet forums and Facebook. Therefore, it cannot be generalized for the whole population. Concerning all those limitations it is obvious that more research is needed to better investigate the real physical activity patterns in women pregnant with twins. Further research on this topic is necessary for obstetric providers to counsel women on appropriate exercise with a twin pregnancy.

## 5. Conclusions

Our study proves that the population of women with a twin gestation is not sufficiently physically active and is often discouraged from performing exercises during gestation although physical activity has no harmful effect of pregnancy outcome and beneficial on pregnancy ailments. Therefore, it is crucial to inform obstetricians and other health care providers to recommend active lifestyle during a twin gestation and to provide easily accessible reliable information on physical activity to pregnant women. More studies concerning multiple pregnancy and physical activity are needed to investigate the impact of particular kinds and extensity of exercises on the course of a twin pregnancy and delivery.

## Figures and Tables

**Figure 1 ijerph-18-07724-f001:**
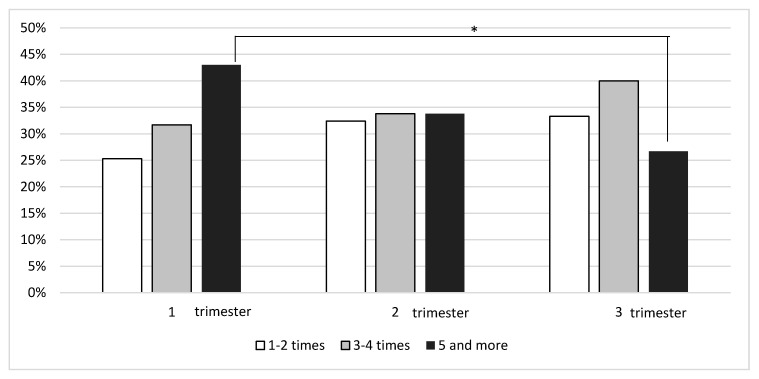
Declared frequency exercises per week in a group of physically active women with a twin pregnancy. * *p* < 0.05.

**Figure 2 ijerph-18-07724-f002:**
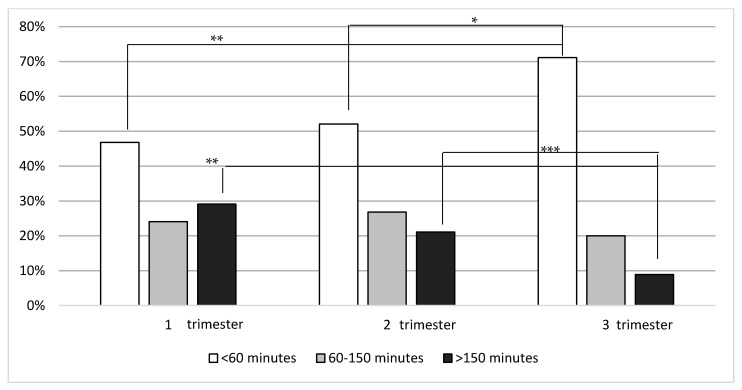
Declared total time of exercises per week in a group of physically active women with a twin pregnancy. * *p* = 0.004; ** *p* ≤ 0.01; *** *p* = 0.017.

**Table 1 ijerph-18-07724-t001:** Characteristics of the study group.

	Study Group *N* = 652	Physically Active *N* = 162	Physically Non-Active *N* = 490	
	% (*N*)	% (*N*)	% (*N*)	*p*
Age (years)				
<20	0 (2)	0 (0)	0 (2)	1.00
21–30	59 (382)	56 (90)	60 (292)	0.41
31–40	40 (264)	43 (70)	40 (194)	0.46
41–50	1 (4)	1 (2)	0 (2)	0.26
Education				
Basic	1 (4)	0 (0)	1 (4)	0.58
Secondary	31 (204)	22 (36)	34 (168)	0.004
Higher	63 (412)	74 (120)	60 (292)	0.001
Vocational	5 (32)	4 (6)	5 (26)	0.53
Inhabitancy				
Countryside	27 (178)	14 (22)	32 (156)	<0.001
Town < 50,000	18 (116)	27 (44)	15 (72)	0.001
City 50,000–100,000	14 (94)	12 (20)	15 (74)	0.44
City 100,000–500,000	18 (114)	22 (36)	16 (78)	0.07
City > 500,000	23 (150)	25 (40)	22 (110)	0.59
Parity				
Primiparous	60 (388)	59 (96)	60 (292)	1.00
Multiparous	40 (264)	41 (66)	40 (198)	1.00
Physical activity before pregnancy				
I didn’t exercise	37 (242)	10 (16)	46 (226)	<0.001
Less than 6 months	13 (82)	6 (10)	15 (72)	0.03
6 months–2 years	22 (144)	28 (46)	20 (98)	0.02
>2 years	28 (184)	56 (90)	19 (94)	<0.001
Pre-pregnancy BMI ^a^				
Underweight	8 (54)	6 (10)	9 (44)	0.32
Normal	61 (398)	67 (108)	59 (290)	0.09
Overweight	21 (136)	20 (32)	21 (104)	0.74
Obese	10 (64)	7 (12)	11 (52)	0.29
Gestational weight gain				
		Normal weight *N* = 398		
adequate	39 (154)	37 (40)	39 (114)	0.73
inadequate	53 (210)	52 (56)	53 (154)	0.91
excessive	8 (34)	11 (12)	8 (22)	0.31
		Overweight *N* = 136		
adequate	43 (58)	44 (14)	42 (44)	1.00
inadequate	44 (60)	44 (14)	44 (46)	1.00
excessive	13 (18)	12 (4)	14 (14)	1.00
		Obese *N* = 64		
adequate	34 (22)	50 (6)	31(16)	0.31
inadequate	53 (34)	50 (6)	54 (28)	1.00
excessive	13 (8)	0 (0)	15 (8)	0.33

^a^ body mass index.

**Table 2 ijerph-18-07724-t002:** Pregnancy ailments in the group of physically active and non-active respondents.

	Study Group *N* = 652	Physically Active *N* = 162	Physically Non-Active *N* = 490	
	% (*N*)	% (*N*)	% (*N*)	*p*
Constipation	43 (280)	41 (66)	44 (214)	0.52
Leg swelling	54 (352)	53 (86)	54 (266)	0.86
Back pain	65 (426)	57 (92)	68 (334)	0.01
Mood swings	45 (296)	47 (76)	45 (220)	0.72
Fatigue	78 (506)	68 (110)	81 (396)	0.001
Sleeping problems	60 (394)	53 (86)	63 (308)	0.03
Decreased libido	33 (212)	32 (52)	33 (160)	0.92
Heartburn	67 (436)	63 (102)	68 (334)	0.25
Leg cramps	36 (234)	31 (50)	38 (184)	0.13
Varicose veins	13 (82)	12 (20)	13 (62)	1.00

**Table 3 ijerph-18-07724-t003:** Delivery and neonatal birthweight in physically active and non-active respondent groups.

	Study Group *N* = 652	Physically Active *N* = 162	Physically Non-Active *N* = 490	
	%/*N*	%/*N*	%/*N*	*p*
Gestational age at delivery (weeks)				
<30	3 (22)	0 (0)	5 (22)	0.002
<34	25 (166)	21 (34)	27 (132)	0.15
<37	67 (436)	60 (98)	69 (338)	0.05
≥37	33 (216)	40 (64)	31 (152)	0.05
GH ^a^/PE ^b^	9 (58)	9 (14)	9 (44)	1.00
ICP ^c^	6 (38)	4 (6)	7 (32)	0.2
GDM ^d^ treated with diet	32 (206)	41 (66)	29 (140)	0.04
Vaginal delivery	7 (48)	11 (18)	6 (30)	0.05
Cesarean section	93 (604)	89 (144)	94 (460)	0.05
Delivery onset	*N* ^e^ 48	*N* ^e^ 18	*N* ^e^ 30	
Spontaneous	79 (38)	67 (12)	87 (26)	0.145
Induced	21 (10)	33 (6)	13 (4)	0.145
First win Dichorionic	*N* = 484	*N* = 120	*N* = 364	
<10th	4 (21)	7 (8)	4 (13)	0.18
<2500 g	51 (248)	55 (66)	50 (182)	0.35
>2500 g	49 (236)	45 (54)	50 (182)	0.35
Second twin Dichorionic				
<10th	7 (36)	8 (9)	7 (27)	1.00
<2500 g	56 (272)	52 (62)	58 (210)	0.29
>2500 g	44 (212)	48 (58)	42 (154)	0.29
First twin Monochorionic diamniotic	*N* = 168	*N* = 42	*N* = 126	
<10th	8 (13)	10 (4)	7 (9)	0.73
<2500 g	71 (120)	52 (22)	78 (98)	0.003
>2500 g	29 (48)	48 (20)	22 (28)	0.003
Second twin Monochorionic diamniotic				
<10th	8 (13)	5 (2)	9 (11)	0.50
<2500 g	68 (114)	62 (26)	70 (88)	0.35
>2500 g	32 (54)	38 (16)	30 (38)	0.35

^a^ Gestational hypertension, ^b^ Preeclampsia, ^c^ Intrahepatic cholestasis of pregnancy, ^d^ Gestational diabetes mellitus, ^e^ Vaginal delivery only.

## Data Availability

Data is available on request.

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
