# Peer review of "Physical Activity Patterns of Women with a Twin Pregnancy—A Cross-Sectional Study"

_ijerph, 2021, doi:10.3390/ijerph18157724_

Round 1
Reviewer 1 Report
Dear authors, I thank you for considering each of the suggestions that I have made. I think your study has been very complete and I thank you for your work.
Author Response
Honourable Reviewer, thank you for your time and all the valuable suggestions.
Reviewer 2 Report
The topic is of high interest in the field of exercise and pregnancy. The reviewers enthusiasm is dampened by the lack of continuity of thought related to the purpose, hypothesis, and results presented.
For example, the purpose is presented as 1) assessing PA patterns, 2) knowledge and experience related to PA with twin pregnancy. However, the hypothesis related to 1) improving fitness (not presented at all), 2) limit GWG (presented in Table 1, but otherwise not mentioned), 3) no adverse outcomes. Then the results focus on 1) PA pattern (this aligns), 2) GDM (could align with adverse outcomes), 3) where they received advice from (not sure what this aligns with), and 4) delivery type (not sure what this aligns with).
Again, the authors need to make sure the question/purpose, hypothesis, and data presented are aligned so that the data is answering the quetsion and testing their hypothesis. This is currently not done in the manuscript.
Author Response
Honourable Reviewer,
Thank you for your time and all the valuable suggestions. The revised the manuscript hypothesis into: “We hypothesize that, analogously to a singleton gestation, only a part of women perform exercises during twin pregnancy although physical activity has no adverse impact on pregnancy outcome”. Therefore, the aims of the study were to: assess the patterns of physical activity during the twin pregnancy, women’s sources of information and experience of women regarding physical activity and its relation to the course of pregnancy and delivery. Our results focus on – physical activity patterns (aim: assess the patterns of respondents’ physical activity during their latest twin pregnancy), course of pregnancy and delivery (like GDM and GWG, aim: relation between physical activity and the course of pregnancy and delivery), sources of information and social attitude (aim: to assess the sources of information and experience regarding physical activity).
Reviewer 3 Report
Dear authors,
thank you very much for efforts to improve your paper.
However, I do believe that pilot-testing a questionnaire is a minimum methodologigal requirement in the application of self-developed questionnaires. Moreover, I checked the questionnaire and I actually doubt that this was the final version of your questionnaire in content. In general, the questionnaire development follows rules in format and content that were broken here.
Moreover, I am still not very happy with your first paragraph as I my opinion it is too similar to the mentioned study from Whitaker et al.
My major problem: I do not understand the dichtomous classification of physically active and inactive when you have questionned physicall activity in very detail. That needs solid explanation and the description of the classification into a dummy variable.
I was very surprised to read that the authors only used data from fully-filled in questionnaires. Never heard about that before in survey research. That's actually not acceptable as you "throw away" relevant data and cause a huge bias, a self-made drop-out bias. From the methodological point of view, this is not acceptable.
Author Response
Honourable Reviewer,
Thank you for your time and all the valuable suggestions.
- However, I do believe that pilot-testing a questionnaire is a minimum methodologigal requirement in the application of self-developed questionnaires. Moreover, I checked the questionnaire and I actually doubt that this was the final version of your questionnaire in content. In general, the questionnaire development follows rules in format and content that were broken here.
The uploaded questionnaire is the English version of the one we used in our survey, however the original one was written in Polish. Unfortunately, we did not perform any pilot-testing of the questionnaire. If you suggest we should do it, we will need to start the study all over again. However, it is likely that we will not gain such a numerous study group.
- Moreover, I am still not very happy with your first paragraph as I my opinion it is too similar to the mentioned study from Whitaker et al.
We have modified the first paragraph again.
- My major problem: I do not understand the dichtomous classification of physically active and inactive when you have questionned physicall activity in very detail. That needs solid explanation and the description of the classification into a dummy variable.
In the study we created special criteria to differentiate physically active women from all study participants. Therefore we divided the population into two groups: physically active marked and inactive. This gave us the opportunity to use the chi-square test to evaluate if there was a statistically significant difference between the two variables. In the survey most questionees could be answered yes or no. Therefore we could asnwet the aims of the study.
- I was very surprised to read that the authors only used data from fully-filled in questionnaires. Never heard about that before in survey research. That's actually not acceptable as you "throw away" relevant data and cause a huge bias, a self-made drop-out bias. From the methodological point of view, this is not acceptable.
The questionnaire was available online and the survey was administered by Google Forms. During completing the survey single consecutive questions were available one by one. Failure to fill in one did not allow the respondents to move on to the next question in the survey. After filling all the questions, they were sent to the server and saved in the database. In on question was not answered, the next one was not available, and the form was not saved. Therefore, only fully-filled questionnaires were analyses.
Round 2
Reviewer 2 Report
Thank you for the thoughtful revisions.
Author Response
Honourable Reviewer,
Thank you for your time and all the valuable suggestions.
Reviewer 3 Report
Dear authors,
thankks for revising the manuscript accordingly.
Author Response
Honourable Reviewer,
Thank you for your time and all the valuable suggestions
This manuscript is a resubmission of an earlier submission. The following is a list of the peer review reports and author responses from that submission.
Round 1
Reviewer 1 Report
The article entitled “Physical activity patterns of over 650 women with a twin pregnancy - a cross-sectional study” addresses a very interesting and little studied topic. Physical exercise during a twin pregnancy.
I agree with each of the limitations that they indicate in their study, they are true, but that does not mean that they are adequately exposed in this article so that it is known in the scientific world.
In the introduction I made less a better exposition of the scientific foundations of the research. Although there is little literature on twin pregnancy and physical exercise, there are studies that speak of the benefits for both the mother and the (single) fetus of physical exercise during pregnancy. I recommend that you improve the theoretical framework of the introduction; physical exercises are known for quality of life, sleep quality, lower weight gain, decreased episiotomy rates with a higher number of eutocic deliveries, lower postpartum depression, etc, ...
Regarding material and methods. They do not describe how these questionnaires were distributed through a website. In the Facebook groups it is not clear to me if it was you who created the Facebook groups or they were already created.
Regarding the exposure of the variables, I observe that it is a bit confused, I think that I could describe each of the blocks of the questionnaire with the variables that are observed in it. And describe how each of the variables will be measured, detailing how they are grouped and why they are grouped in that way. In its methodology, the block is described in a generic way and then variables appear in the results and the discussion that had not been described in the methodology.
The statistical analysis reflected in the methodology is not complete, not all the analyzes to be carried out have been defined, in the discussion it speaks of correlations and it is not expressed in the methodology.
Potential sources of bias have also not been specified.
Results. In this section I observe that the first paragraph begins by discussing with a previous investigation of yours, that corresponds to the discussion section and not to the results section.
Table 3 shows the incorrect categorization of the variables, that is, it does not establish intervals between the variables encompassing each other. In the variable Gestational age at delivery (weeks) he establishes four categories, but the category <37 as expressed would encompass the previous categories and I do not understand why he expresses it this way or what he is looking for with it.
I point out the same as in the methodology section. It is necessary to describe the variables analyzed in the methodology so that they are better understood in the results.
In the Discussion, lines 220 to 222 describe exercise during pregnancy and before pregnancy, although there are no data on twin pregnancies, these rates can be compared with single pregnancies, especially before pregnancy.
On line 226 he talks about a study of his own, so that readers do not believe that it is a self-quote, I recommend that you also cite other published studies found in the scientific literature.
On line 283 he describes significant correlations that are not described in the methodology.
In conclusion, it would be important to describe the beneficial or non-harmful effects observed in the study for the woman and the fetus during pregnancy and delivery.
Finally, in the declaration of informed consent, he writes that the informed consent of all the subjects involved in the study and their consent to publish this article had been obtained. However, in the methodology he points out that the ethics committee did not consider it necessary, I beg you to clarify this aspect.
Reviewer 2 Report
Title: The title is succinct, however, the “over 650” can be deleted.
Abstract: This is clearly written and the conclusions, in general, support the data. Although, OBs should recommend for women with multifetal pregnancies to be active, this study was not designed to determine the safety and level of exercise for these women. Therefore, the concluding statements should be made with caution. Thus, further research on this topic is necessary in order for obstetric providers to counsel women on appropriate exercise with a twin pregnancy.
Introduction:
- The first sentence should refer to exercise during pregnancy rather than exercise in general.
- Rather than just ACOG, other guidelines from an international perspective should be mentioned (e.g. Obstet Gynecol Surv. 2014 Jul; 69(7): 407–414. doi: 1097/OGX.0000000000000077)
- The pregnancy complications listed at the end of the intro, pg 2 of 12, are not necessarily addressed by gestational exercise (e.g. monochorionicity, IUGR).
- Although the authors clearly state the purpose is “to assess the knowledge and experience of women regarding PA during their latest twin pregnancy,” the hypothesis states that “exercise during a twin pregnancy could improve fitness, limit GWG, and improve general condition.” Thus, the purpose and hypothesis are not parallel. Is the study about 1) women’s knowledge and experience or 2) whether exercise in a twin pregnancy improves their fitness, limits GWG, and improves their overall health, or 3) both of these?? This needs to be clarified.
Methods:
- The outcomes from the participant questionnaire are clearly outlined as follows: 1) sociodemographic (age, education, inhabitancy), 2) a) exercise recommendation provided to them and 2)b)where they looked for PA during pregnancy info , 3) PA patterns during pregnancy (type, frequency, duration each week), 4) ailments during pregnancy, and 5) their opinions on family attitudes about exercising while pregnant with twins.
- The cutoff for “physically active during pregnancy” is well below (e.g. it could be as low as only 30 minute per week, with no threshold for intensity) the standard cutoff of 150 minutes of moderate intensity exercise, therefore, this either needs to be justified for twin pregnancy or the cutoff altered. A very low cutoff will lead to misinterpretation of the data, since the threshold is technically a fairly inactive person.
- The methods mention cutoffs for BMI, GWG, PTD, neonatal birth weight, however, it does not state how this data was collected.
Results:
- The study population begins with type of twin pregnancy, however, how this information was obtained is not clearly stated in the methods section.
- Beginning with table 1 aligns with the presentation of part 1 of the questionnaire, but then the information skips section 2a, 2b, to discuss PA patterns during pregnancy.
- Table 1 presents data as 2 groups (physically active vs. physically non-active), then Figure 1 presents 3 group comparison (based on frequency 1-2X vs. 3-4X vs. 5+X per week), whereas Figure 2 presents a different 3 group comparison (based on minutes <60min vs. 60-150min vs. 150+ min per week). This should be streamlined to have one type of group comparison that aligns with the purpose and hypothesis.
- So far, Table 1, Figures 1-2, and the first 3 paragraphs of results do not present data to answer either the knowledge/experience nor the potentially improved fitness/GWG/overall health of the women as the study initially proposed in the intro/purpose/hypothesis.
- Section 3.3 and Table 2 is the first presentation of data that aligns with the proposed hypothesis related to improving general condition during pregnancy.
- Section 3.4 and Table 3 present data related to pregnancy outcomes between the 2 groups of physically active vs. physically non-active. Although this is important information to begin to determine the influence of exercise during twin pregnancy, the variables are not described in methods nor do they align with either the purpose nor hypothesis statements.
- Section 3.5 presents data related to pregnancy outcomes related to questionnaire section 2a) with the group as a whole but the variables do not align with either the purpose (women’s knowledge/experience) nor the hypothesis statements (potentially improved fitness/GWG/overall health of the women).
- Section 3.6 presents data related to social attitudes about exercise during pregnancy, related to questionnaire section 5 with the group as a whole but the variables do not align with either the purpose (women’s knowledge/experience) nor the hypothesis statements (potentially improved fitness/GWG/overall health of the women).
Discussion:
- The discussion begins by stating the importance is that this is the 1st study to investigate patterns of PA in women with twin pregnancy, however, this was neither the purpose nor the hypothesis initially presented.
- The discussion is difficult to critique since the significant findings cannot be ascertained without a clear purpose and hypothesis that are aligned throughout the intro, methods, analysis, and results sections.
Conclusion:
- Similarly, the conclusion cannot be critiqued since the significant findings cannot be ascertained without a clear purpose and hypothesis that are aligned throughout the intro, methods, analysis, and results sections.
- The authors state their study “proves that the population of women with a twin gestation is not sufficiently physically active” and “is often discouraged from performing exercises during gestation,” though this was not the purpose, hypothesis, nor how the data was analyzed.
Tables & Figures:
- Participant characteristics presented in Table 1 mostly align with section 1 from the participant questionnaire. However, the methods do not mention how some of this information was obtained, such as parity, PA before pregnancy, prepregnancy BMI, and GWG. Please add this detail in the methods section.
- Table 2 data presents a 2 group comparison that clearly demonstrates 3 significantly improved ailments of general health related to being PA during a twin pregnancy.
Reviewer 3 Report
- Comments to Author
Physical activity patterns of over 650 women with a twin preg-2 nancy – a cross-sectional study.
Overview and general recommendation:
The authors investigated the relevant topic of knowledge and experience of 652 women regarding physical activity during their latest twin pregnancy. The manuscript addresses an interesting public health issue.
However, the manuscript has severe weaknesses.
The introduction needs to be revised completely as it is very similar to the Whitaker et al stud. The methods section lacks diverse relevant information and the conduction of the analyses remains unclear.
We explain our concerns in more detail below, divided into.
The manuscript needs to be edited in language.
2.1 Major Comments:
Abstract:
- The abstract lacks important methodological information, e.g. whether a validated questionnaire was used, clarification of dependent and independent variables, etc.
Introduction:
- One major problem is that authors do not discuss the current state of recommendations, e.g. ACOG suggests that women who carry multiples refrain from aerobic exercises.
- There are exercises that are recommended, just as walking. Here authors should distinguish between the type of activities and check available guidelines.
- For multiple pregnancies also see “Canadian Guideline for Physical Activity throughout Pregnancy”.
- What I really did not like: The first paragraph is like “content copy paste” from the Whitaker et al. study. https://link.springer.com/article/10.1186/s12884-019-2574-2
For comparison: Whitaker et al.
The twin birth rate has risen nearly 80% over the last four decades in the United States, accounting for 1 in every 30 births in 2016 [1]. Compared to singleton pregnancies, women pregnant with twins are at greater risk for adverse pregnancy outcomes, including hypertensive disorders, gestational diabetes, anemia, postpartum hemorrhage, operative delivery, uterine rupture, and prolonged hospitalization [2, 3]. Twin pregnancies are also associated with a 4- to 10-fold increased risk of perinatal morbidity and mortality compared to singleton pregnancies, largely driven by the increased risk of preterm birth, low birth weight, and intrauterine growth restriction [2, 3]. There are many non-modifiable risk factors that account for the disproportionate morbidity experienced in twin gestations. However, appropriate physical activity and proper nutrition during pregnancy are increasingly recognized as important modifiable factors that contribute to maternal and child outcomes [4].
Physical activity in pregnancy is associated with minimal risks and has been shown to provide health benefits to most women. The American College of Obstetricians and Gynecologists (ACOG) recommends all women with uncomplicated pregnancies engage in 20–30 min of moderate-intensity aerobic physical activity on most or all days of the week [4]. These recommendations are consistent with the U.S. Department of Health and Human Services (HHS) Physical Activity Guidelines for Americans, which state that women should do at least 150 min of moderate-intensity aerobic activity a week during pregnancy [5]. Health care providers who see pregnant women (e.g., obstetricians, midwives, nurse practitioners) are advised to carefully evaluate women with medical or obstetric complications before making recommendations on physical activity participation. There are no specific physical activity guidelines for women pregnant with twins; however, given the higher risk for maternal complications in twin gestations, it is important that health care providers evaluate the risks and benefits of physical activity for each patient and counsel accordingly.
Methods
- The methods section is a “mix of all” and lacks structure.
- It is recommended to divide the method section into subsections e.g. setting, sampling, Data collection, Questionnaire statistical analysis etc. for a better overview.
- Clarify dependent and independent variables, confounding variables, etc.
- Clarify the single analysis.
- In the methods section, it is not apparent whether they used a validated questionnaire to collect data. This point is only explained in more detail in the discussion. Please also add this point in the methods section.
- Was the questionnaire pilot-tested? Did the questionnaire receive saturation in pilot-testing?
- I suggest adding the questionnaire as an additional file to the document – as a reviewer I would like to see the questionnaire.
- I find the gradation of the main variable in Physically active und Physically non-active is very hard. Important information and intermediate aspects are lost as a result. I classify the assessment of the Physically of the two groups as difficult.
- How did the authors deal with missing values? Please complete.
Results
- All in all, many tables and figures lead to the fact that it quickly becomes confusing. I would advise to use fewer tables and focus on the relevant information as specifically information from graphics can be captured in one sentence.
- However, it remains still unclear what is the “real focus” of the paper. This reflects the lack of clarity and precision in the methodological section.
- Changing the steps of the percentage bar in figure 1 and figure 2 can lead to confusion, please make it consistent.
- Unfortunately there are only descriptive analysis and authors did not investigate research questions applying more complex analyses. Therefore, a “real research questions” also lacks.
Discussion
- Overall, the discussion of results is very long and the discussion of methods is rather short. I recommend to adjust the parts
- It might be helpful to shorten to relevant/ new / or surprising results in the discussion of results It is not necessary that all results be discussed.
- In the methods discussion should be pointed out that further studies in longitudinal design are important to test the results for stability and causality and of course, the research question may adapted to investigate associations, causslities etc.
Since informed consent was not obtained directly, as described in the methods section, I recommend revising the following note: “Informed Consent Statement: Informed consent was obtained from all subjects involved in the study. Informed consent has been obtained from the patient(s) to publish this paper.”
2.2 Minor Comments
Methods
- For future studies I recommend to ask socio-demographic data only at the end of a self-developed questionnaire. The literature shows that this increases the response rate.
- Some information do not belong to the methods section, e.g. that there is no evidence-based information on proper weight gain for underweight women in twin pregnancies – that rather belongs to the introduction